# Relationships between Serotonin Transporter Availability and the Global Efficiency of the Executive Control Brain Network

**DOI:** 10.3390/ijms25115713

**Published:** 2024-05-24

**Authors:** Rémi Janet, Edmund Derrington, Jean-Claude Dreher

**Affiliations:** UMR5229, Neuroeconomics Laboratory, CNRS-Institut de Sciences Cognitives Marc Jeannerod, 69500 Lyon, France; janet.remi@ymail.com (R.J.); edmund.derrington@univ-lyon1.fr (E.D.)

**Keywords:** fMRI-PET, serotonin, executive control network

## Abstract

The diverse effects of serotonin on cognition may emerge from the modulation of large-scale brain networks that support distinct cognitive processes. Yet, the specific effect of serotoninergic modulation on the properties of these networks remains elusive. Here, we used a simultaneous PET-fMRI scanner combined with graph theory analyses to investigate the modulation of network properties by the Serotonin Transporter (SERT) availability measured in the dorsal raphe nucleus (DRN). We defined global efficiency as the average mean of efficiencies over all pairs of distinct nodes of specific brain networks, and determined whether SERT levels correlated with the global efficiency of each network. SERT availability in the DRN correlated negatively with the global efficiency of the executive control brain network, which is engaged in cognitive control and directed attention. No relationship was observed between SERT availability and the global efficiency of the default mode or the salience brain networks. These findings indicate a specific role of serotoninergic modulation in the executive control brain network via a change in its global efficiency.

## 1. Introduction

The serotoninergic system plays a pivotal role in influencing various aspects of our daily behaviors by shaping large-scale brain networks [1,2]. In particular, it is associated with attention regulation, cognitive control and learning, emotional and motivational processes and mood regulation [3,4,5,6,7,8]. The serotonin reuptake transporter (SERT), which is located at the pre-synaptic site, is strategically situated to play a key role in the regulation of the large networks involved in such processes [9,10]. Yet, little is known about how serotoninergic activity modulates the properties of these networks, to ultimately influence cognition.

The fluctuation of cognitive states may be accounted for by the modulation of large brain networks overlapping with the default mode network, the executive control brain network, or the salience network [11]. For example, the salience network (SN) selectively recruits brain regions that are needed to appropriately adapt our behavior when facing a particular situation. This network can be considered as a rapid and short-term adaptative network. In contrast, the executive control network (ECN) functions in the making of goal-oriented behaviors and longer-term planning to achieve a specific goal [12]. The executive control network supports attention and cognitive flexibility [12,13,14,15]. It has been proposed that stress triggers a reallocation of neural resources toward the salience network, which supports rapid but rigid decisions, at the expense of the executive control network, which supports flexible decisions [16]. Here, we tested whether serotoninergic activity in the dorsal raphe nucleus (DRN) modulates such functional connectivity differences within or between brain regions at the scale of large brain networks.

To test this hypothesis, we investigated whether SERT levels in the DRN correlate with the characteristics and integrity of large-scale brain networks. We used graph-theory analysis to estimate the functional connectivity of brain networks and measure their global efficiency. Global efficiency appears to be a good marker to investigate how well information is exchanged over such brain networks [17]. High global efficiency is related to greater exchanges of information. We measured both BOLD fMRI and [^11^C]-DASB, a highly selective radioligand for visualization of the serotonin transporter (SERT) in vivo, using a novel simultaneous PET-fMRI scanner. We combined this multimodal neuroimaging approach with measures of the global efficiency of the salience, the default mode, and the executive control brain networks using the graph theory (GT) analysis. We used predefined networks [18] and the functional connectivity toolbox CONN [19] to estimate the modulation of the global efficiency by the free SERT level in the DRN (Figure 1). Given that the serotoninergic neurons in the DRN have widespread projections in cortical regions, we expected to find a covariation between DRN SERT levels and the global efficiency of these brain networks.

## 2. Results

We first defined the networks using pre-established regions of interest from which we extracted the functional connectivity metrics using the graph theory framework [18]. We defined three networks (Figure 1A), known as the default mode network (DMN, in red in Figure 1A), the executive control network (ECN, in green in Figure 1A), and the salience network (SN, in yellow in Figure 1A). All regions of interest included in each of these networks will be defined as nodes in our graph theory analysis. Next, we extracted the SERT level in the DRN as a proxy of the overall serotoninergic activity (Figure 1B). We then investigated whether the SERT level extracted from the DRN correlated with the efficiency of each network composed of the different ROIs.

### 2.1. Higher Executive Control Network Efficiency in Individuals with Lower SERT

We investigated a putative modulatory effect of SERT levels in the DRN on the connectivity of each of the three predefined functional networks by applying graph theory analysis. Our analysis revealed no significant modulation of the global efficiency of either the DMN (*p* = 0.07) or the SN (*p* = 0.85). However, the global network efficiency in the ECN showed a statistically significant negative correlation with SERT levels in the DRN (*p* = 0.002, beta = −0.07). Thus, a lower level of SERT availability in the DRN was associated with greater global efficiency in the ECN (Figure 2A; Table 1). Similarly, we observed no statistically significant modulation of either the degree (*p* = 0.1 and *p* = 0.41) or path length (*p* = 0.58 and *p* = 0.77) metrics in the DMN and SN, respectively. However, although we observed no significant modulation of path length in the ECN, reduced levels of SERT in the DRN correlated with a significant increase in the degree metric (*p* = 0.004, β= −0.29) (Figure 2B; Table 1).

### 2.2. Right Inferior Parietal Gyrus Drives the Effect

We further analyzed the modulation of the global network efficiency at each individual node of the ECN to establish the importance of each node in determining global efficiency throughout the whole network. Only the right inferior parietal gyrus (*p* = 0.019 FDR corrected, β = −0.22) (Figure 3; Table 2) demonstrated an apparent significant modulation of the global efficiency correlated with DRN SERT levels, with higher SERT availability in the DRN associated with a lower global efficiency within this brain region. No significant modulation of either degree or path length was observed for any other node in the ECN network.

## 3. Discussion

The level of free SERT in the DRN is thought to reflect the general level of serotoninergic activity. Therefore, we studied how free SERT levels in the DRN correlate with functional connectivity in large scale brain networks. This may allow us to elucidate how perturbation of serotonergic activity affects behavior and ultimately quality of life [20,21]. It can also inform us on the interindividual differences in response to treatments targeting the serotoninergic system. Here, we focused on three brain networks: the default mode network, the executive control network, and the salience brain network using a simultaneous PET-fMRI scanner to measure SERT levels. We analyzed network connectivity using graph theory analysis and found that higher levels of free SERT in the DRN were associated with lower global efficiency in the executive control brain network. However, we observed no link between SERT level in the DRN and any connectivity metrics of the salience or default mode brain networks. When investigating the degree of connectedness and the path length of the executive control network, we found that only the degree of connectedness, and not the path length, was modulated by the level of free SERT in the DRN (Figure 2). Furthermore, the only specific brain region in the ECN subject to this modulation was the right inferior parietal gyrus (Figure 3). The DRN has previously been shown to be functionally integrated into the ECN [22]. Our results show that SERT availability in the DRN modulates the global efficiency of the executive control network, and especially that of the inferior parietal network. Moreover, graph properties revealed that the average number of connections was lower when SERT DRN availability was high in the ECN. This is the first time that network connectivity has been investigated as a function of the SERT availability in the DRN.

In contrast, our findings indicate no modulation of the SN or the DMN by SERT availability in the DRN. This suggests that, within the healthy population, the SN and the DMN network connectivity, and thus information exchanges within these networks, are not strongly modulated by DRN SERT levels. In contrast, studies conducted with depressed participants have revealed decreased connectivity between the DRN and prefrontal regions as well as the anterior cingulate cortex [23,24]. These studies identified the potential for the serotoninergic system to modulate large-scale network integrity. Other studies that investigated the effect of serotonin levels on the connectivity of the DRN (using both acute tryptophan depletion and supplementation) found no modulation of connectivity between brain regions and the dorsal raphe nucleus [25]. In contrast to the current experiment, these authors used pharmacological interventions to modulate serotonin levels and studied the functional connectivity of the DRN with the brain. Here, we investigate how the functional properties of the networks covaries with the serotoninergic activity as measured by SERT DRN level. Previous studies that investigated how SERT availability varies as the result of acute tryptophan depletion show that SERT levels fall after such depletion in non-human primates [26]. It is possible that the ratio between serotonin and SERT availability is more important for serotoninergic signaling than the level of serotonin per se.

The global efficiency of the ECN appears to be subject to the global efficiency of connectivity within the right inferior parietal gyrus. This brain region includes part of the angular gyrus and the posterior supramarginal area and is involved in attentional processes, and is also a part of the sensorimotor processing loop [27,28,29,30]. The parietal cortex, and especially its right side, is thought to integrate memorized features as a multimodal conscious representation that enables subjective re-experiencing of previous events, and mediates the allocation of top-down attention to specific aspects of the environment, depending on the subject’s goals [31]. The increased global efficiency of connectivity within the right inferior parietal gyrus, when SERT availability in the DRN is low, suggests an increase in processing in this region compared to when SERT availability is high. Thus, when SERT availability is low or when SERT is inhibited, individuals might be expected to more accurately integrate and maintain multiple information sources to guide behavior.

We acknowledge several limitations to our study. First, functional networks were defined according to a previous parcellation of the brain [18]. This could lead to a less specific definition of the different networks compared to a data-driven approach. In addition, extending this study to a larger sample that includes women might be useful to identify any sex-related bias in our analysis. Indeed, it has been shown that men have higher SERT levels than women [32]. Similarly, replicating the analysis on a larger sample that covers a broader age-range might allow us to clarify whether the relationship we observed still holds in different age groups. A previous study has shown a significant reduction of SERT binding with age in humans [33]. Not only SERT levels are modulated with age. It has been shown that small-world properties of brain connectivity as well as executive functioning are modulated with increasing age [34,35]. Finally, due to the relatively small sample size, results should be considered as a preliminary investigation and replication should be conducted, especially regarding the small effects we discovered. To go further, analyses of brain segregation could be conducted to determine whether serotonin modulates the relative integration of regions from a whole-brain perspective. In addition, the use of Dynamic Granger Causality might allow us to characterize the dynamic effective on connectivity, to provide a better understanding of the contribution of the serotoninergic system to these network efficiency transitions.

To conclude, our results suggest that increased serotonin activity, evidenced by lower levels of free SERT, improves ECN network efficiency. This result suggests a possible role of serotoninergic transmission in the modulation of how information is exchanged within this network and could account for how well individuals integrate the information and consequently adapt their behavior. Our study not only contributes to a deeper understanding of serotonin’s impact on cognition but also suggests potential implications for targeted therapeutic interventions in cognitive disorders.

## 4. Methods

### 4.1. Participants

Thirty healthy male volunteers, aged 19 to 32 years (mean 23.4 ± (SD) 2.9) were recruited via the University Claude Bernard Lyon 1 mailing list. For inclusion, participants met the following criteria: native French-speaking, right-handed, no current medical treatment, no history of neurological or psychiatric disorders, and no auditory, olfactory, or visual deficits. All volunteers were screened further for general MRI counter-indications. A physician conducted medical examinations for inclusion criteria such as physical and psychological health. Participants gave their written consent and received monetary compensation for the completion of the study. This study was approved by the Medical Ethics Committee (CPP Sud-Est IV, ID RCB: 2016-A01588-43).

### 4.2. PET and MRI Acquisition Were Performed Simultaneously in a Siemens Biograph mMR

#### 4.2.1. MRI Data Acquisition

All MRI acquisitions were performed on a 3 Tesla scanner using EPI BOLD. We performed the scans on a PET-MRI Biograph mMR Siemens at the CERMEP Bron (single-shot EPI, TR/TE = 2000/25, flip angle 85°, 39 axial slices interlaced with 3 mm thickness gap, FOV = 192 × 192 × 129). We collected a total of 300 volumes over the entire session, in an interleaved ascending manner. Acquisition speed was increased by applying an accelerator factor of 2 and a Grappa mode. We waited for stabilization of the signal to acquire the first scan.

Whole-brain high-resolution T1-weighted structural scans consisted of a 3D sagittal T1-weighted sequence. The anatomical volume covered the entire brain using slices with a 1 mm thickness (1 × 1 × 1 mm) (repetition time = 2300 ms; echo time = 2.34 ms; flip angle = 8; field of view = 350 × 263 × 350 mm; voxel size = 1 × 1 × 1 mm^3^).

#### 4.2.2. PET Data Acquisition

PET acquisition started with an intravenous injection of a bolus of [11C]-DASB, a radiotracer that binds SERT. Acquisitions were performed in list-mode over 90 min. Subsequently, the PET data underwent list mode motion correction [36], followed by rebinning into 24 time frames of variable lengths (8 × 15 s, 3 × 60 s, 5 × 120 s, 1 × 300 s, 7 × 600 s) for dynamic reconstruction. The reconstruction of images utilized OP-OSEM 3D, incorporating the system point spread function with parameters set at 3 iterations of 21 subsets. Sinograms underwent correction for scatter, random events, normalization, and attenuation [37]. Reconstructions were executed with a zoom factor of 2, which resulted in a voxel size of 1.04 × 1.04 × 2.08 mm^3^ within a matrix of 344 × 344 × 127 voxels. Lastly, Gaussian post-reconstruction filtering (FWHM = 2 mm) was applied to the PET images.

#### 4.2.3. PET Preprocessing and Kinetic Modeling

For co-registration purposes, an average PET image was computed and the anatomical T1 MPRAGE was registered to this average PET image (using rigid transformation). We used the Hammersmith 83 regions atlas to label regions of the brain [38,39]. Regional time activity curves were then extracted on the subject space after having co-registered the atlas map on the subject space. We then used a Simplified Tissue Reference Model (SRTM) and the cerebellar grey matter (assumed to be devoid of SERT transporters) as a reference region to compute parametric images of non-displaceable binding potential (BPND). Finally, we normalized the resulting PET images to the standard Montreal Neurological Institute (MNI) atlas space using the DARTEL (diffeomorphic anatomical registration through exponentiated lie algebra) toolbox procedure [40] (using the T1 SPM template and resulting in voxels of 2 × 2 × 2 mm).

### 4.3. Functional Resting State Data Preprocessing

Image analyses were performed using SPM12 (Wellcome Department of Imaging Neuroscience, Institute of Neurology, London, UK, http://fil.ion.ucl.ac.uk/spm/software/spm12/, accessed on 1 May 2020). Time-series images were registered in a 3D space to minimize any effects that might result from head movements. Once DICOMs were imported, functional scans were corrected for slice timing and realigned to the first volume. Structural images were co-registered on the average computed dynamic PET image. In this way, we ensured that both the fMRI and PET images were in the same space. Finally, to make group and individual comparisons, EPI images were co-registered with structural maps and normalized spatially into the standard Montreal Neurological Institute (MNI) atlas space using the DARTEL procedure (the Diffeomorphic Anatomical Registration Through Exponentiated Lie algebra) [40] (using the T1 SPM template and resulting in voxels of 2 × 2 × 2 mm, according to the same procedure as the PET parametric images). Images were then smoothed spatially using an 8 mm isotropic full-width at half-maximum (FWHM) Gaussian kernel using standard procedures in SPM12.

The resting-state fMRI (rs-fMRI) images were implemented in the Functional Connectivity toolbox CONN, [19]. This allowed us to perform denoising of the resting state functional images. This step consists of removing factors that are identified to be potential confounding effects for the estimated BOLD signal. Such effects were estimated and removed separately for each voxel and for each subject and functional session, using the Least Squares Method (LSM). The first factor employed was noise related to the physiological parameters. The use of principal component analysis on the noise of a region of interest (white matter and cerebrospinal fluid) added as nuisance regressors in a GLM significantly reduces the temporal standard deviation in resting scans [41]. Furthermore, the signal from head movements and the primary derivative of such movements have also been found to be confounding factors [42]. Images were finally filtered for frequency bands between 0.008 and 0.09 Hz [43].

### 4.4. Network ROI Definition

The DMN, ECN, and SN networks, as identified by a previous study, were used as predefined networks [18]. The DMN concerned 19 regions, which included the medial prefrontal cortex, the posterior cingulate cortex and the bilateral angular gyrus. The SN contained 18 regions, including the bilateral anterior insula, the anterior cingulate cortex, the bilateral superior prefrontal cortex, and the bilateral supramarginal gyri. The ECN comprised 12 regions, including the bilateral superior frontal gyrus, middle frontal gyrus, angular cortex, right thalamus, and right caudate nucleus. It also included the bilateral part of the cerebellar cortex. The Appendix A provides complete network definitions (Figure 1A). Finally, the SERT level was extracted from the DRN region according to the anatomical region defined in the AAL3 atlas [44] (Figure 1B).

### 4.5. Functional Connectivity Matrices at the Single Subject Level

In order to investigate the connectivity of brain networks and their modulation by SERT levels from the DRN, we first conducted a task-weighted analysis at the single subject level and created the connectivity matrices. We used a bivariate regression analysis and weighted the signal according to the hemodynamic response function, which represents the functional connectivity between regions in our predefined networks. This allowed us to investigate the correlations between our regions of interest during the second level analysis.

### 4.6. Graph Theory Analysis

Initial studies in network neuroscience revealed that brain networks on various spatial scales have properties similar to other biological and non-biological networks, such as the “small-world” property. In order to perform graph theory analysis, and identify the covariation of the SERT level in the DRN with each network’s global efficiency, we used the analysis path implemented in the CONN toolbox. A graph is defined as a set of nodes (vertices) linked by connections (edges) and provides an abstract representation of interactions between a system’s elements. To construct the individual adjacency matrix, we applied a z-scored transformation to the Pearson correlation coefficient and kept only positive elements that had a correlation coefficient above 0.85 (z-scored).

First of all, we investigated global efficiency at the network level or node level. The global efficiency of a network represents the overall efficiency of the information transfer (using a *p*-value of *p* < 0.05 uncorrected). Global efficiency of a node represents the average inverse distance between a specific node and all the other nodes in the graph [17]. To test for significant effects at the node level, we used a *p*-value of 0.05 corrected for the False Discovery Rate (FDR).

We investigated also two other different metrics of graph theory that influence global efficiency. The first is the degree metric. The degree of a node represents the number of connections established with other nodes in the network. For a network, it represents the average number of connections of each node within the network. The second metric we used was the path length. The path length of a node is defined as the average shortest-path distance between this node and all other nodes in the subgraph of connected nodes. The path length of a network represents the measure of interconnectedness of the entire network. Shorter path lengths are considered to be more desirable.

## Figures and Tables

**Figure 1 ijms-25-05713-f001:**
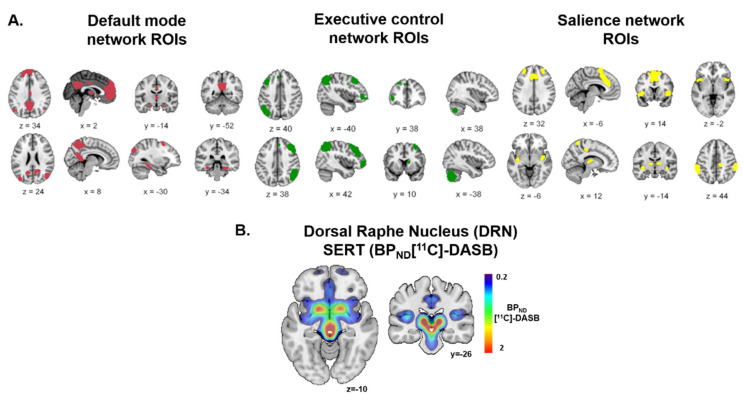
(**A**). Definition of the default mode, executive control, and salience brain networks. We used predefined regions included in the default mode (red), executive control (green), and salience networks (yellow) as nodes in our graph theory analysis. Adapted from W. Shirer et al., 2012 [18] (https://findlab.stanford.edu/functional_ROIs.html, accessed on 1 May 2020). (**B**). Dorsal raphe nucleus in white and the SERT level (non-displaceable Biding Potential BP_ND_ [^11^C]-DASB).

**Figure 2 ijms-25-05713-f002:**
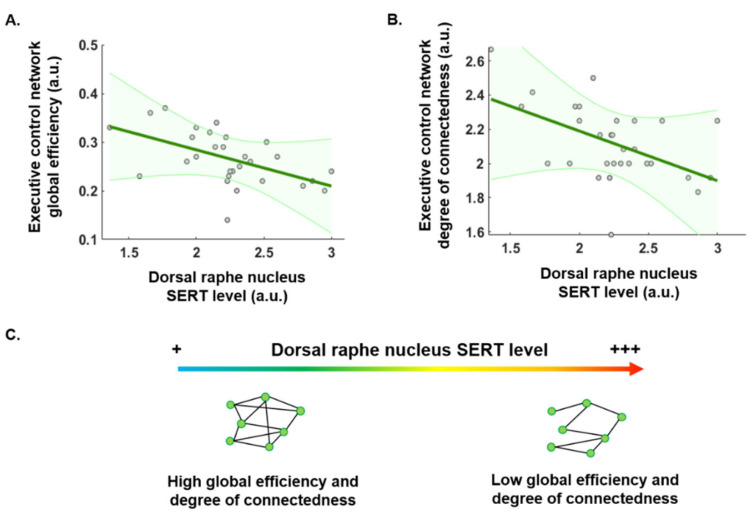
Modulatory effect of SERT availability. Correlation between graph theory metrics (global efficiency and degree) linking the entire executive control network (ECN) (on the left side). (**A**). Negative correlation between the global efficiency of the ECN and the SERT level in the DRN. (**B**). Negative correlation between the average degree of connectedness of the ECN network and the SERT level in the DRN. Analyses at the network level were significant at *p* < 0.05. (**C**). Illustration of the modulation of global efficiency and the degree of connectedness of the ECN according to the level of free SERT in the dorsal raphe nucleus. The green shaded areas represent the confidence interval of the correlation. The green circle surrounded by grey represent individual data. + corresponds to a low SERT level and +++ corresponds to a high SERT level. SERT = serotonin reuptake transporter.

**Figure 3 ijms-25-05713-f003:**
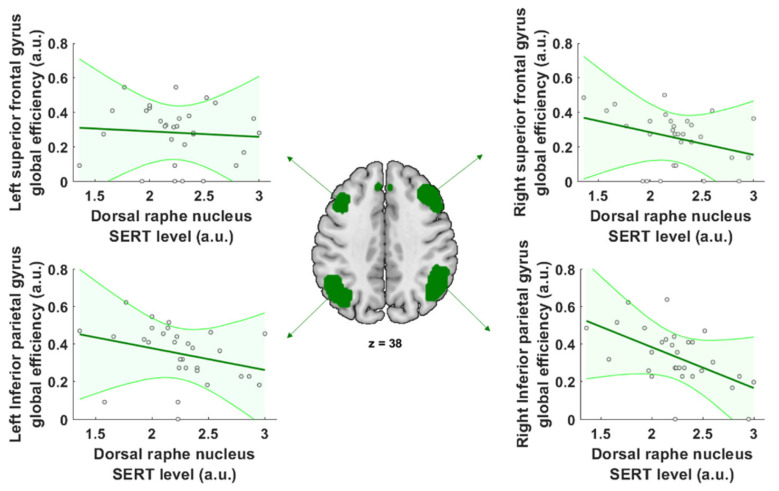
Regional modulation of global efficiency by SERT. Correlation plots between SERT level in the DRN and the global efficiency of the left superior frontal gyrus (top left plot), the left inferior parietal gyrus (bottom left plot), the right superior frontal gyrus (top right plot), and the left inferior frontal gyrus (bottom right plot). Only the right inferior parietal gyrus regional global efficiency negatively correlates with the SERT level in the DRN (*p* = 0.019 FDR corrected). The green shaded areas represent the confidence interval. The green regions represent ROIs used to defined nodes of the ECN. They correspond to the networks’ ROIs define in Figure 1A. Both SERT levels and global efficiency are measured in arbitrary units (a.u.). The grey circles represent individual data points. The green lines correspond to the correlation slope. The green arrows show from which brain regions the correlation plots are derived.

**Table 1 ijms-25-05713-t001:** Network metrics moderated by the level of SERT in the dorsal raphe nucleus. Higher free SERT in the dorsal raphe nucleus is related to lower global efficiency and lower degree of connectedness in the executive control network (ECN). No effect was observed for the default mode network (DMN) or salience network (SN). n.s. = not significant.

		DMN		ECN		SN	
		β	*p*-unc	β	*p*-unc	β	*p*-unc
Network	Global efficiency		n.s.	−0.07	0.003		n.s.
	Average path length		n.s.	n.s.	n.s.		n.s.
	Degree		n.s.	−0.29	0.005		n.s.

**Table 2 ijms-25-05713-t002:** Regional metrics of the executive control network modulated by the SERT level in the dorsal raphe nucleus. Higher free SERT in the dorsal raphe nucleus is related to lower global efficiency in the right inferior parietal gyrus regions of executive control network (ECN). No effect was observed for the default mode network (DMN) or salience network (SN) ROIs. n.s. = not significant.

	DMN			ECN			SN		
	β	*p*-unc	Region	β	*p*-unc	Region	β	*p*-unc	Region
Global efficiency		n.s.		−0.22	0.019	Right inferior parietal gyrus		n.s.	
Average path length		n.s.			n.s.			n.s.	
Degree		n.s.			n.s.			n.s.	

## Data Availability

Data are available upon request to Jean-Claude Dreher at dreher@isc.cnrs.fr.

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
