# Peer review of "Relationships between Serotonin Transporter Availability and the Global Efficiency of the Executive Control Brain Network"

_ijms, 2024, doi:10.3390/ijms25115713_

Round 1

Reviewer 1 Report

Comments and Suggestions for Authors

1. The manuscript contains numerous grammatical errors in Section 4.3 and other sections, resulting in low readability. It is recommended to carefully revise the manuscript to enhance its readability.

2. The manuscript uses inconsistent terminology, such as "figure" and "Figure", "graph theoretical analysis" and "graph theoretical approach". It is recommended to use a consistent terminology throughout the manuscript.

3. Is the Connectivity toolbox the same as the CONN toolbox?

Comments on the Quality of English Language

Moderate editing of English language required.

Reviewer 2 Report

Comments and Suggestions for Authors

By using pre-established definition of salience and executive brain networks and the graph theory metrics, Janet and collaborators investigated the modulation of functional connectivity by the level of the free serotonin reuptake transporter (SERT) from the dorsal raphe nucleus, as a way to assess the overall serotoninergic activity. The submitted manuscript is concise and well-argued, but in general I think it is structured in such a way that it is not accessible to a wide readership. I believe that, as it is presented, the potential of what has been reported can only be fully understood by a small number of experts in the field.

Here below, few comments and suggestion

Line18-20 – Please rephrase. The reported findings may suggest a novel role of serotoninergic modulation, as it is reported in the Discussion section, considering that results should be taken as preliminary due to the small (and sex-biased) sample size.

Line 29 - Remove the reference Hariri et al., 2002

Line 33 – Add a reference

Line 35 – please check the reference, is it correct?

Line 45-47 – Authors should explain further why they focused on this hypothesis.

Results – It is weird in my opinion that Fig. 1 is missing in the Results section. Perhaps the information used for the precise network definition and SERT levels extraction (lines 271-274) should be included in a paragraph at the beginning of the results section. Please decide whether the data depicted in Fig.1 are supplementary or main data (line 271).

Fig. 2A, B – Please define what the green shaded area is in the caption

Fig. 3 – Please define what the green shaded area is in the caption. Please explain what the green colored region in the reported brain transversal section are in the caption. Do they refer to Fig. 1?

Lines 169-178 – The authors correctly underline the limitations of the work in the Discussion section, including the small number of samples. They do not mention the rather narrow age range analyzed (understandable considering the system used for recruiting) and the fact that the selected participants are males only. Both choices should be justified or discussed in the manuscript. Is an age dependent effect excluded? May the analyses be biased by a sex-effect?

Round 2

Reviewer 1 Report

Comments and Suggestions for Authors

The authors have addressed the comments made by the reviewers in the revised version of the manuscript. Therefore, I have no further comment.